Analysis of genetic diversity among Chinese Cyclocybe chaxingu strains using ISSR and SRAP markers

Liu Jin-Hao 1 2 3
Ding Fang-Hui 2
Song Hai-Yan 1 4
Chen Ming-Hui 1 2 3
Hu Dian-Ming hudianming1@163.com 1 2 3
1 Jiangxi Agricultural University, Bioengineering and Technological Research Centre for Edible and Medicinal Fungi , Nanchang , China
2 Jiangxi Agricultural University, College of Bioscience and Bioengineering , Nanchang , China
3 Jiangxi Key Laboratory for Conservation and Utilization of Fungal Resources, and College of Bioscience and Bioengineering , Nanchang , China
4 Ministry of Education of the P.R. China, Key Laboratory of Crop Physiology, Ecology and Genetic Breeding (Jiangxi Agricultural University) , Nanchang , China
Khan Mudassar
Electronic publication date: 2022 Sep 29
Publication date: 2022
Volume: 10
Electronic Location ID: e14037
Received 2022 May 19; Accepted 2022 Aug 18
Copyright: ©2022 Liu et al.
Copyright year: 2022
Copyright holder: Liu et al.
License: This is an open access article distributed under the terms of the Creative Commons Attribution License, which permits unrestricted use, distribution, reproduction and adaptation in any medium and for any purpose provided that it is properly attributed. For attribution, the original author(s), title, publication source (PeerJ) and either DOI or URL of the article must be cited.
License URL: https://creativecommons.org/licenses/by/4.0/

Keywords: Cyclocybe chaxingu, Germplasm resource, Genetic diversity, Molecular markers

Funding: National Natural Science Foundation of China NSFC 32070023 NSFC 32060014 Key Projects of Youth Fund of Jiangxi Science, Technology Department of China 20192ACBL21017 Natural Science Foundation of Education Department of Jiangxi Province of China GJJ190168 Jiangxi Agriculture Research System Funds for research were provided by the National Natural Science Foundation of China (NSFC 32070023, NSFC 32060014), the Key Projects of Youth Fund of Jiangxi Science, Technology Department of China (20192ACBL21017), the Natural Science Foundation of Education Department of Jiangxi Province of China (GJJ190168) and the earmarked fund for jiangxi Agriculture Research System. The funders had no role in study design, data collection and analysis, decision to publish, or preparation of the manuscript.

==============================
Background

Cyclocybe chaxingu is an edible and medicinal fungal species commonly cultivated in China. The major problems currently facing by growers of C. chaxingu is the random labeling of strains and strains aging and degeneration. Therefore, an evaluation of genetic diversity is essential for the conservation and reproducing programs of this species.

Methods

In the present study, 24 widely cultivated strains were collected from the main producing areas of China, and the genetic diversity analysis was performed. DNA polymorphism among these Chinese C. chaxingu strains was analyzed using inter-simple sequence repeat (ISSR) and sequence-related amplified polymorphism (SRAP) markers.

Results

Eight ISSR primers amplified a total of 75 DNA fragments of which 61 (81.33%) were polymorphic. Fifteen SRAP primer combinations amplified 166 fragments of which 132 (79.52%) were polymorphic. Cluster analysis showed that the C. chaxnigu strains fall into five groups with a genetic distance values ranging from 0.06 to 0.60 by ISSR analysis, while the SRAP analysis divided the test strains into four groups within the range of genetic distance from 0.03 to 0.57. The results of the present study reveal a high level of genetic diversity among the widely cultivated C. chaxingu strains.

Introduction

Cyclocybe chaxingu is an edible and medicinal species, which commonly cultivated in Jiangxi, Yunnan and Fujian provinces of China  (Vizzini, Claudio & Ercole, 2014; Liu et al., 2021b; Wang et al., 2016). C. chaxingu is chosen by consumers for its nutritional properties such as high protein, low fat and sugar, and its pharmacological effects such as anti-oxidation, anti-aging, and anti-tumor properties  (Wu et al., 2019; Lee et al., 2010; Lee et al., 2009). It is cultivated on a large scale and the estimated output of C. chaxingu in 2019 was almost 0.9 million tons in China  (Chinese Edible Fungi Association, 2022).

Depends on its properties such as independence of environmental parameters and the high levels of detectable polymorphism, DNA molecular markers has played an indispensable role in biological germplasm identification and innovation (Paterson, Tanksley & Sorrells, 1991; Bian & Song, 2005). In previous studies, inter-simple sequence repeat (ISSR), and sequence-related amplified polymorphism (SRAP) techniques have been widely used in the analysis of genetic diversity. ISSR uses semi-arbitrary markers amplified by PCR in the presence of one primer complementary to a target microsatellite  (Zietkiewicz, Rafalski & Labuda, 1994). SRAP uses two sets of positive and negative primers to amplify the Open Reading Frame (ORF), including the intron and promoter region (Li & Quiros, 2001). These two markers have previously been utilized for the genetic diversity analysis of many mushrooms species such as Auricularia auricula-judae (Yao et al., 2018), Auricularia polytricha (Yu et al., 2008), Lentinula edodes (Liu et al., 2015), Lepista nuda  (Du et al., 2018), Pleurotus citrinopileatus (Zhang et al., 2012), Pleurotus eryngii (Aref et al., 2018) and Pleurotus pulmonarius (Yin et al., 2014). Genetic diversity analysis using combined ISSR and SRAP markers has been proven to be reliable and effective (Tang et al., 2010).

Breeding fine strains of high yield and high quality is a primary task to meet the demand of the rapid development of modern C. chaxingu industry. Germplasm is the basics of C. chaxingu breeding. In this study, combined ISSR and SRAP markers were adopted for 24 strains of C. chaxingu cultivated in China to analyze the genetic diversity and the relationships. This information will facilitate the efficient program for the purpose of breeding and conservation.

Materials & Methods

Mushroom strains

A total of twenty-four strains of C. chaxingu strains were collected from the main producing areas of China (Table 1), which were deposited in the Culture Collection of Jiangxi Agricultural University (JAUCC), were used in this study. A previous study based on ITS, SSU and RPB2 phylogenetic analysis showed that all the test strains belonged to C. chaxingu (Liu, 2020).

Table 1 Cultivated strains of C. chaxingu used in this study.

No.	Strain No.	Source	No.	Strain No.	Source	
1	JAUCC 0727	JXAUa	13	JAUCC 2119	HAUf	
2	JAUCC 1842	NCJb	14	JAUCC 2184	HAU	
3	JAUCC 1847	LCJc	15	JAUCC 2189	MIJSg	
4	JAUCC 1850	LCJ	16	JAUCC 2192	SDh	
5	JAUCC 1851	LCJ	17	JAUCC 2195	SD	
6	JAUCC 1918	FJd	18	JAUCC 2196	SD	
7	JAUCC 1920	YNe	19	JAUCC 2205	JXAU	
8	JAUCC 1921	YN	20	JAUCC 2206	SCi	
9	JAUCC 1922	YN	21	JAUCC 2207	JXAU	
10	JAUCC 1925	YN	22	JAUCC 2214	JLj	
11	JAUCC 1926	YN	23	JAUCC 2924	YN	
12	JAUCC 1927	YN	24	JAUCC 4926	YCJk	
Notes.

a Jiangxi Agricultural University.

b Ningdu County, Jiangxi Province.

c Lichuan County, Jiangxi Province.

d Derived through tissue isolation from mushroom growing farm in Fujian Province.

e Derived through tissue isolation from mushroom growing farm in Yunnan Province.

f Huazhong Agricultural University.

g Microbiological Institute of Jiangsu Province.

h Derived through tissue isolation from mushroom growing farm in Shandong Province.

i Derived through tissue isolation from mushroom growing farm in Sichaun Province.

j Derived through tissue isolation from mushroom growing farm in Jilin Province.

k Yingtan County, Jiangxi Province.

DNA extraction

The mycelia grew on PDA at 25 °C for 8 days were used to extract DNA for molecular marker analysis. Genomic DNA was extracted from 100 mg of dry mycelia using a modified cetyltrimethyl ammonium bromide (CTAB) method (Huang, Ge & Sun, 2000; Doyle & Doyle, 1987). The quality and quantity of the genomic DNA were determined with a NanoDrop 2000 spectrophotometer (Thermo Scientific, Waltham, MA, USA) and electrophoresis on a 1.0% agarose gel. DNA samples were diluted to 50 ng/µl for PCR amplification.

ISSR and SRAP analysis

A total of 23 primers/primer pairs (Biomed, Beijing, China) that produced clearly distinguishable and reproducible fragments were selected and used in this study for ISSR and SRAP analysis (Table 2). All of the amplification reactions were performed in a PCR Amplifier (Bio-Rad T100TM Thermal Cycler; Bio-Rad, Hercules, CA, USA) in 25-µL reaction mixtures.

Table 2 Primer sequences used for ISSR and SRAP analyses in this study.

			SRAP primers	
	ISSR primers (5′–3′)	Annealing temperature (°C)	Forward primers (5′–3′)	Reverse primers (5′–3′)	
P1	TGCACACACACACAC	54	me1	TGAGTCCAAACCGGATA	em1	GACTGCGTACGAATTAAT	
P2	GTGACACACACACAC	54	me2	TGAGTCCAAACCGGAGC	em2	GACTGCGTACGAATTTGC	
P3	GTGACGACTCTCTCTCTCT	55	me3	TGAGTCCAAACCGGAAT	em3	GACTGCGTACGAATTGAC	
P4	GGATGCAACACACACACAC	55	me6	TGAGTCCAAACCGGTAG	em4	GACTGCGTACGAATTTGA	
P10	GAGAGAGAGAGAGAGAC	51			em5	GACTGCGTACGAATTAAC	
P12	AGAGAGAGAGAGAGAGGC	55			em6	GACTGCGTACGAATTGCA	
P22	AAGAAGAAGAAGAAGAAGC	46			em7	GACTGCGTACGAATTATG	
P23	GAGAGAGAGAGAGAGACT	53			em8	GACTGCGTACGAATTAGC	

For ISSR analysis, the reaction mixtures contained 12.5 µL of 2 × Taq PCR Master Mix (Vazyme, Nanjing, China), 1 µL of primer (10 µM/L), 1 µL of template DNA, and 10.5 µL of ddH2O. Amplification program was: 4 min of denaturing at 94 °C, 35 cycles of 35 s at 94 °C, 45 s at 46–55 °C (see Table 2 for primer annealing temperature), 2 min at 72 °C and followed by a final extension of 10 min at 72 °C.

For SRAP analysis, the reaction mixtures contained 12.5 µL of 2 × Taq PCR Master Mix (Vazyme, Nanjing, China), 1 µL of each primer (10 µM/L), 1 µL of template DNA, and 9.5 µL of ddH2O. The amplification included an initial denaturation at 94 °C for 5 min, 5 cycles of 94 °C for 1 min, 35 °C for 1 min and 72 °C for 1 min, followed by 35 cycles of 1 min at 94 °C, 1 min at 50 °C, and 1 min at 72 °C, and a final extension of 10 min at 72 °C.

Amplified products were fractionated by electrophoresis in 2% (w/v) agarose/TAE gels, visualized under UV after staining with TS-Gelred (Tsingke Biotechnology Co., Ltd., Beijing, China), and documented using a gel documentation and image analysis system (GenoSens 2000; CLiNX, Shanghai, China).

Data analysis for genetic diversity

Qualitative scoring of bands was done from gel photographs obtained from ISSR and SRAP analysis with “1″for presence and “0″for absence to generate a binary matrix. Only those bands amplified consistently were considered. Smeared and weak bands were excluded from the analysis. For data structures with only 0 and 1, Jaccard similarity coefficient is generally adopted (Ivchenko & Honov, 1998). The genetic distances were estimated based on Jaccard similarity coefficient. A cluster analysis was performed based on the genetic distances using the Unweighted Pair Group Method of Arithmetic Average (UPGMA) by R Statistical Software (R Core Team, 2021) and vegan package (Oksanen et al., 2020). The goodness of fit of the clustering to the data matrix was calculated. And the optimal grouping strategy is to select the number of groups corresponding to the maximum average contour width (Rousseeuw, 1987). Genetic diversity analysis was performed using the POPGENE program (version 1.32) (Yeh et al., 1999). The number of amplified loci (N), the percentage of polymorphic loci (PPL), the number of effective alleles (Ne), the Nei’s gene diversity (H) and the Shannon information index (I) were calculated for each primer and among all primers (Nei, 1973; Lewontin, 1972). In addition, a Principal Coordinate Analysis (PCoA) (Longya, Talumphai & Jantasuriyarat, 2020) was performed using the R Statistical Software to obtain a graphical representation of the relationship between the 24 test genotypes.

Results

In total, eight ISSR and 15 SRAP primers or primer pairs gave reproducible results that were further considered for data analysis. Table 3 shows the total number of bands and the percentage of polymorphisms for each primer or primer pair.

Table 3 Characteristics of ISSR and SRAP markers used in this study.

Primers	N a	NPL b	PPL (%) c	Ne d	H e	I f	
P1	9	9	100	1.628	0.355	0.527	
P2	7	7	100	1.638	0.374	0.555	
P3	6	5	83.33	1.686	0.373	0.533	
P4	7	5	71.43	1.496	0.285	0.418	
P10	13	10	76.92	1.549	0.313	0.457	
P12	10	6	60	1.343	0.210	0.319	
P22	12	9	75	1.379	0.211	0.321	
P23	11	10	90.91	1.657	0.375	0.546	
me1 + em1	10	9	90	1.647	0.367	0.536	
me1 + em3	9	7	77.78	1.386	0.235	0.362	
me1 + em4	11	6	54.55	1.344	0.200	0.297	
me1 + em5	16	15	93.75	1.582	0.343	0.511	
me2 + em3	10	6	60	1.511	0.273	0.387	
me2 + em6	8	5	62.5	1.444	0.252	0.368	
me2 + em7	12	10	83.33	1.592	0.337	0.493	
me3 + em2	12	10	83.33	1.595	0.335	0.489	
me3 + em4	11	9	81.82	1.705	0.376	0.533	
me3 + em5	9	7	77.78	1.564	0.318	0.463	
me3 + em6	18	17	94.45	1.604	0.336	0.498	
me3 + em7	12	11	91.67	1.535	0.308	0.455	
me3 + em8	11	7	63.64	1.395	0.235	0.351	
me6 + em1	9	8	88.89	1.403	0.244	0.373	
me6 + em3	8	5	62.5	1.289	0.175	0.264	
Total	241	193					
Average	10.48	8.39	80.08	1.521	0.297	0.437	
ISSR average	9.38	7.63	81.33	1.547	0.312	0.459	
SRAP average	11.07	8.8	79.52	1.506	0.289	0.425	
Notes.

a Number of total loci.

b Number of polymorphic loci.

c Percentage of polymorphic.

d Effective number of alleles.

e Nei’s gene diversity.

f Shannon information index.

Genetic diversity based on ISSR marker

A total of 24 primers were initially screened to produce polymorphic patterns and only 8 of them were selected which gave reproducible and distinct polymorphic amplified products. For representational purposes, the extent of polymorphism revealed by primer P10 is shown in Fig. 1. The data collected from inter-simple sequence repeat (ISSR), detected total of 75 loci in 24 strains, out of which 61 (81.33%) were polymorphic, with an average of 9.38 polymorphic fragments per primer (Table 3). The ISSR primer P1 and P2 gave the highest polymorphism (100%), while the lowest polymorphism (60%) was detected by the P12 primer. The values of Ne, H and I were 1.547, 0.312 and 0.459, respectively.

Figure 1 Representative ISSR amplification profile using primer P10.

For the 24 C. chaxingu strains, the genetic distances estimated based on Jaccard coefficient using ISSR data (see Table S1) varied from 0.06 (JAUCC 2192 and JAUCC 2196) to 0.60 (JAUCC 0727 and JAUCC 1927), with an overall mean of 0.40. The co-phenetic correlation for the ISSR dendrogram was estimated at 0.93, corresponding to a good fit. A dendrogram constructed from the Jaccard distances matrix using the UPGMA method was shown in Fig. 2A. All the test strains were grouped into five main clusters by calculating the maximum average contour width value. Cluster I and Cluster III each comprised a single genotype (JAUCC 0727 and JAUCC 1927), while Cluster II, Cluster IV and Cluster V were delineated into two sub-clusters. Within Cluster II, Cluster IV and Cluster V, JAUCC 1925, JAUCC 2119 and JAUCC 1926 appeared to be distinct from the other genotypes, respectively. Interestingly, most strains cultivated in Yunnan Province were clustered together at the distance level of 0.45. Groupings identified by UPGMA analysis were confirmed by PCoA data (Fig. 2B). The two most informative PCoA components accounted for 44.21% of the variation observed.

Figure 2 Cluster analysis and PCoA analysis based on ISSR data.

(A) UPGMA dendrogram of 24 C. chaxingu strains constructed using Jaccard distance analysis based on molecular profiles revealed by ISSR marker; (B) 2D principal component analysis (PCoA) based on genetic distance from ISSR data using R Statistical Software.

Genetic diversity based on SRAP marker

Among the 48 SRAP primer pairs tested in the study, 15 primer pairs were further used to characterize C. chaxingu strains. A representative set of amplification profiles obtained with primer combination me3 + em7 is shown in Fig. 3. The present study showed that out of 166 loci, 132 (79.52%) loci were polymorphic showing an average of 11.07 polymorphic loci per primer pairs tested (Table 3). The maximum polymorphic loci were generated by primer pairs em6+me3 (94.45%). Among the primers pairs studied, primers pairs em6+me3 generated the highest 18 loci, while primer combination em6+me2 and em3+me6 generated the lowest eight loci. The values of Ne, H and I were 1.506, 0.289 and 0.425 based on SRAP marker.

Figure 3 Representative SRAP amplification profile generated using the primer combination me3 + em7.

With the cluster analysis on SRAP molecular marker, the two most closely related strains were found to be JAUCC 1847 and JAUCC 1851 (genetic distance was 0.03), and the two most distantly related strains were JAUCC 1847 and JAUCC 2924 with lowest similarity index (genetic distance was 0.57). Genetic distance estimated based on Jaccard coefficient obtained by SRAP profile with an average value of 0.39. Cluster analysis of SRAP data (see Table S2) based on the distance matrix generated a dendrogram with four major groups in a maximum average contour width value (Fig. 4A). The co-phenetic correlation for the SRAP dendrogram was estimated at 0.96, which showed a strong goodness of it. Within Cluster I and Cluster III, JAUCC 1918 and JAUCC 0727 were each comprised a single genotype. In the SRAP analysis, strain JAUCC 0727 showed a closer relationship with the other strains than in ISSR analysis. Similarly, there was a tendency to get together within most strains form Yunnan Province (JAUCC 1920, JAUCC 1921, JAUCC 1922, JAUCC 1925, JAUCC 1927), at genetic distance level of 0.38. Principal coordinate analysis (PCoA) data based on the genetic distance matrix are shown in Fig. 4B. These revealed similar groupings to UPGMA, and confirmed the genetic uniqueness of genotypes JAUCC 0727. The two most informative PCoA components accounted for 63.81% of the variations observed.

Figure 4 Cluster analysis and PCoA analysis based on SRAP data.

(A) UPGMA dendrogram of 24 C. chaxingu strains constructed using Jaccard distance analysis based on molecular profiles revealed by SRAP marker; (B) 2D principal component analysis (PCoA) based on genetic distance from SRAP data using R Statistical Software.

Genetic diversity based on combined ISSR and SRAP markers

A total of 23 primers/primer pairs were used for the analysis of combined ISSR and SRAP data, genetic distance among all the test strains ranged from 0.06 (JAUCC 2192 and JAUCC 2196) up to 0.54 (JAUCC 1851 and JAUCC 1920). A total of 241 loci in 24 strains, out of which 193 (80.08%) were polymorphic, with an average of 10.48 polymorphic fragments. The co-phenetic correlation for the combined ISSR and SRAP dendrogram was estimated 0.97, corresponding to a very good fit. Dendrogram by using UPGMA and Jaccard coefficient grouped the 24 test strains into five main clusters with a maximum average contour width value (Fig. 5A). Within Cluster IV, JAUCC 0727 appeared as a single genotype once again. Strains cultivated in Yunnan Province clustered together (genetic distance was 0.4) same as using ISSR or SRAP molecular marker alone.

Figure 5 Cluster analysis and PCoA analysis combined ISSR and SRAP data.

(A) UPGMA dendrogram of 24 C. chaxingu strains constructed using Jaccard distance analysis based on molecular profiles revealed by combined ISSR and SRAP markers; (B) 2D principal component analysis (PCoA) based on genetic distance of combine ISSR and SRAP data using R Statistical Software.

The PCoA based on ISSR and SRAP data revealed that the strains belonging to a particular cluster were grouped together in the PCoA plot (Fig. 5B). Groupings identified by UPGMA analysis and Jaccard coefficient were confirmed by PCoA data which also revealed that the strain JAUCC 0727 was genetically very distinct from the other genotypes. The two most informative PCoA components accounted for 61.01% of the variation observed.

Discussion

As one of the most important edible fungi in China, the research on cultivation of C. chaxingu is relatively intensive, but there is a lack of research on genetic diversity. Therefore, efforts have been made in the present study to characterize twenty-four strains of C. chaxingu collected from different parts of China, using inter-simple sequence repeat (ISSR) and sequence-related amplified polymorphism (SRAP) markers.

The host and geographical diversity were the root causes of the genetic diversity of C. chaxingu. Wild strains of C. chaxingu mostly occurs on decayed and dead wood of broadleaf trees such as Camellia oleifera, Populus spp. and Salix spp. It is mainly distributed in temperate and subtropical regions of China, such as Jiangxi, Fujian, Hunan, Sichuan and Yunnan Provinces  (Liu et al., 2021a). In the previous study, the genetic diversity of eight C. chaxingu varieties in Hunan Province was analyzed using amplified fragment length polymorphism (AFLP) technique, the percentage of polymorphic loci amplified by AFLP primers (Bao et al., 2021) was 94.10% higher than characterized by ISSR and SRAP (81.33% and 79.52%) in this study. It could be that ISSR and SRAP techniques target different parts of the genome. Although the percentage of polymorphic loci amplified by AFLP primers is higher, it is cumbersome and technically difficult to operate, prefer to choose molecular markers that are easier and cheaper in this study, such as ISSR and SRAP. Since each DNA maker system has its own advantages, it is important to use more than one DNA marker system in the analysis of genetic diversity (Aref et al., 2018; Longya, Talumphai & Jantasuriyarat, 2020). ISSR, SRAP and ISSR + SRAP dendrograms of the 24 test strains generally exhibited highly similar clustering patterns. For example, strains JAUCC 1920, JAUCC 1921, JAUCC 1922, JAUCC 1925 and JAUCC 1927 from the Yunnan Province in China clustered together in each case. The strain JAUCC 0727 which collected from the campus of Jiangxi Agricultural University always form a separate branch and more closely related to strains from Yunnan. These results possibly indicating that cultivated strains in Yunnan, with a narrow genetic basis, had been domesticated from wild-type strains. In fact, the findings indicated that Intra-strain ITS heterogeneity with positional double peaks was identified in most C. chaxingu strains (Liu, 2020; Qian et al., 2021), suggested that most of the strains cultivated in China may have been obtained through hybridization. We learned that there are a lot of farmers who were using the random labeling of strains and the introduction into different regions of identical strains under different designations to grow C. chaxingu for economic benefit, especially in Jiangxi and Fujian provinces. The lack of new germplasm and the haphazard-introduction of species from one region to another may impede breeding efforts. Therefore, explore more wild resources and breeding good character, high yield and stable C. chaxingu strains is conducive to the development of C. chaxingu industry. The success of strain selection depends on the investigation of genetic diversity (Otieno et al., 2015).

DNA maker system is an efficient tool to reveal genetic relationship among different genotypes through the numbers of polymorphisms detected  (Sikdar et al., 2010). Surveying the genetic variation through ISSR and SRAP analysis could be useful in the selection of parental strains for breeding purposes. In the present study, based on ISSR and SRAP markers, ISSR marker produced a higher percentage of polymorphic bands (81.33%) than SRAP marker (79.52%). However, the average number of polymorphic bands from different markers detected by SRAP primers was 11.07, which was higher than that of ISSR primers (9.38). The dendrograms from the ISSR, SRAP, and ISSR + SRAP combined data displayed slightly different groupings. Rather than selecting one molecular marker, we prefer to combine two molecular markers for genetic diversity analysis. Mixed multiple marker systems provide more complete genome information. The wild strain JAUCC 0727 is far related to other strains in the three groups of classification results. Wild strains of C. chaxingu should be introduced into breeding programs to increase the genetic heterogeneity and to improve the commercial characters of the cultivated strains. In summary, this study laid a foundation for genetic improvement and breeding of C. chaxingu.

Supplemental Information

Table S1 ISSR data

Click here for additional data file.

Table S2 SRAP data

Click here for additional data file.

Supplemental Information 1 Code

Click here for additional data file.

The authors would like to thank the members of Bioengineering and Technological Research Centre for Edible and Medicinal Fungi of Jiangxi Agricultural University (Nanchang, China) for their technical support.

Additional Information and Declarations

Competing Interests

Author Contributions

Data Availability

The authors declare there are no competing interests.

Jin-Hao Liu conceived and designed the experiments, performed the experiments, analyzed the data, prepared figures and/or tables, and approved the final draft.

Fang-Hui Ding performed the experiments, prepared figures and/or tables, and approved the final draft.

Hai-Yan Song conceived and designed the experiments, authored or reviewed drafts of the article, and approved the final draft.

Ming-Hui Chen analyzed the data, authored or reviewed drafts of the article, and approved the final draft.

Dian-Ming Hu conceived and designed the experiments, analyzed the data, authored or reviewed drafts of the article, and approved the final draft.

The following information was supplied regarding data availability:

The ISSR data and SRAP data and the code are available in the Supplementary Files.

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
