# Peer review of "Analysis of genetic diversity among Chinese Cyclocybe chaxingu strains using ISSR and SRAP markers"

_PeerJ, doi:10.7717/peerj.14037_

## Round 0.1 · original submission · Major Revisions

Dear authors,

Thank you for submitting to PeerJ. The Reviewers have suggested improvements (Major and Minor) that you need to address. One Reviewer has suggested Major while the other has suggested Minor revisions. Based on your reply and improvement, the manuscript will be judged further.

Thanks.

Mudassar Nawaz Khan
Handling Editor

·

Basic reporting

The authors have studied the genetic diversity among Chinese Cyclocbe chaxingu strains using PCR based ISSR and SRAP genetic markers with the purpose to select markers/method for precise identification and classification of commercial C. chaxingu strains for Chinese and overseas markets. Consequently, the author has analyzed genetic distances and has performed cluster analysis using the UPGMA. Furthermore, genetic diversity and Principal Coordinate Analysis (PCoA) has been performed using the POPGENE and the R Statistical Software to obtain a graphical representation of the relationship between the 24 strains.
My comments on the paper go into some detail, but the primary weakness of the paper is technical language and grammar.

Experimental design

No comment

Validity of the findings

No comment

Additional comments

The paper establishes significant genetic diversity among the C. chaxingu strains but the key objective as mentioned “precise identification and classification of commercial C. chaxingu strains for Chinese and overseas markets” has not been fulfilled. The study could be of significance for the breeding and genetic improvement of C. chaxingu by utilizing the genetic diversity in the tested strains. However, author need to incorporate morphological and biochemical data and find out correlation between genetic markers, morphological and biochemical parameters.

Reviewer 2 ·

Basic reporting

The English language of the manuscript is very poor. I suggest to take help of a colleague who is proficient in English or contact a professional editing service.

Experimental design

Experimental design of the study is good.

Validity of the findings

A suitable rationale relevant to the current study should be added at the end of introduction.
In conclusion section suggest markers which you found more useful based on the current study.

Annotated reviews are not available for download in order to protect the identity of reviewers who chose to remain anonymous.

---

## Round 0.2 · accepted · Accept

Dear Authors,
I am happy to inform you that your manuscript submitted to PeerJ was accepted. Please go through the formal procedures as per journal policy.

Sincerely,
Dr. Mudassar Nawaz Khan

·

Basic reporting

The paper is organized and presented well. The abstract is clear and understandable, introduction could have been elaborated but it is fine and covers the relevant literature, methods is clear, results and discussion are well explained and interpreted with relevant literature. The language has been improved.

Experimental design

No comment

Validity of the findings

No comment

Additional comments

The authors have studied the genetic diversity among 24 widely cultivated strains of Chinese Cyclocbe chaxingu from the main areas of China by applying ISSR and SRAP markers. Consequently, they performed DNA extraction, PCR and cluster analysis to find the genetic diversity. The paper establishes significant genetic diversity among the selected strains of Chinese Cyclocbe chaxingu.
Previously, I had suggested to change the main objective of the study and that has been done accordingly. Moreover, adding morphological and biochemical parameters could have improved the significance of the study but the authors rightly explained the reason why that has not been added.
This paper should be accepted for publication in its current form.

Reviewer 2 ·

Basic reporting

no comment

Experimental design

no comment

Validity of the findings

no comment

Additional comments

no comment